# Analysis of the applicability and utility of a gamified didactics with exergames at primary schools: Qualitative findings from a natural experiment

**Alejandro Quintas-Hijós[1], Carlos Peñarrubia-Lozano[2], Juan Carlos Bustamante[3]***

**1** Department of Education Sciences, University of Zaragoza, Huesca, Spain, **2** Department of Musical, Plastic and Corporal Expression, University of Zaragoza, Zaragoza, Spain, **3** Department of Psychology and Sociology, University of Zaragoza, Zaragoza, Spain

* jbustama@unizar.es

**Data Availability Statement:** All relevant data are within the paper and its Supporting Information files.

## Abstract

One of the main objectives of Physical Education in elementary schools is to encourage motivation so that the subject enhances academic performance and the practice of physical exercise. Didactic research should evaluate the effectiveness of educational methods to know if they are applicable, useful, and in what sense. Exergames are digital motor games that aim to stimulate players' motor skills. Gamification refers to the use of game-based elements in nongame contexts to motivate actions. This research evaluates a gamified exergaming intervention, designed to improve children's academic performance by focusing on understanding applicability and usefulness. A natural experiment was set up in schools according to a mixed methods design. The qualitative data herein reported were collected during a natural experiment with a nonrandomized controlled design. The qualitative research design was used with field notes, an open-questions questionnaire, individual semi-structured interviews and focus group interviews. Eight teachers and 417 students took part. A content analysis was chosen as the methodological orientation. The facilitators were the realism of their didactic design and their adaptability to different educational contexts. The main barriers were the required materials and facilities. Teachers and students' attitudes were very positive, although future use was inconclusive. These findings may imply that this study is one of the few to provide positive evidence for educational gamification. The "Mechanics-Dynamics-Aesthetics" gamification model and the "Just Dance Now" exergame may be applicable and useful for didactics in Physical Education, but all the participants' suggestions need to be considered to improve teaching interventions.

## Introduction

Innovative educational interventions should have scientific support to explain whether students learn more and better. Exergames are digital motor games that aim to stimulate players' motor skills, which are popular on the global market and have been paid increasing attention by educational research [1, 2]. As a game type, exergames can be applied to Physical Education

**Funding:** This research has been supported by Grants received by JCB from the Fundación Hergar (2018/0081) (https://www.fundacionhergar.org/), from the Fundación Bancaria Ibercaja (JIUZ-2017-SOC-06) (https://www.fundacionibercaja.es/), and from FEDER (Programa Operativo 2014-2020) and the Regional Aragón Government (INFR2016_UZ_SOC_05) (https://www.aragon.es/-/feder-fondos-fondos-europeos-gobierno-de-aragon). Finally, the first author (AQH) wishes to thank the Spanish Ministry of Education, Culture and Sport (http://www.mecd.gob.es/) for the funding used to obtain a predoctoral university contract to conduct this study. The funders had no role in study design, data collection and analysis, decision to publish, or preparation of the manuscript.

**Competing interests:** The authors have declared that no competing interests exist.

(PE), and can simultaneously provide the benefits of motor games [3] and video games [4]. Gamification refers to the use of game-based elements in nongame contexts for motivating actions [5, 6]. While gamification is advancing in business or marketing, its application to education is still an emerging practice [7]. There are several implicit assumptions about the usefulness of gamification in the education context, and the main one is that gamification is motivating. However, as the scientific literature is still scarce and inconclusive, more empirical studies are needed [8].

As most children in Europe and the USA play video games [9], the question is no longer whether children play video games and how we can prevent them from doing so, but how we can positively impact what type of digital games they use, and why they are playing [10].

Exergames have often been applied to PE in an instrumental and mechanical way [11]. The inclusion of new technology in class does not mean an essential change in didactics. Following Freinet [12], it is necessary to transform the dynamics of the whole classroom to make an essential change in didactics [12]. This contextualized learning change can be created by the gamification strategy [13]. Research in the fields of exergames [2, 14, 15] and gamification [6, 16–18] has shown that both can produce beneficial psychological effects.

One of the main objectives of PE is to encourage motivation toward the subject to increase academic performance or the practice of physical exercise (PEx) [19]. It has been suggested that exergames should be incorporated into PE classes to enhance students' motivation [20]. Six components that exergames posit have been proposed to intrinsically motivate children to engage in PE: challenge, control, curiosity, constant feedback, creativity, competition [21]. Besides, the following motivation-related game characteristics have been postulated: fun, challenging, motivating, developmental, appropriate, individualized, contemporary [22]. Several psychological benefits of exergaming associated with motivation have already been proven, such as providing changes toward physically active behavior [23], especially in those who do not normally engage in traditional PEx [15], by improving either motivation thanks to social interaction during exergames [24] or situational interest motivation [25] by improving academic performance in subjects other than PE [26]. However, empirical support for the effectiveness of exergaming on students' motivation and in-class activity in PE is sparse [25]. Even a review study on exergames suggests that perhaps strengths and positive effects are more likely to be published than null or even negative effects. Thus empirical research that focuses specifically on potential negative effects is needed [10].

Qualitative research has shown that intervention and context could be improved to achieve more success and greater effectiveness by detecting events as distractions for children by social interactions, angered by errors in exergames, or lack of space to correctly perform the activity [27]. Another qualitative study has shown that, despite not quantitatively detecting increased physical activity (PA) through intervention, qualitative results suggest that sufficient exergaming had not been used during the intervention to make any significant differences [28]. All these research works have proven the advantage of using mixed methods to completely interpret empirical research.

According to children, the likelihood of long-term engagement with exergames depended on game content and children's ages (more appropriate for younger children than for teenagers). According to parents, the cost of exergames and lack of space at home to play games were identified as barriers [29]. A positive attitude was found toward the use of exergames in PE in elementary students, but no associations were found between that attitude with gender, physically active habits and digital leisure habits [30]. Another qualitative study found children and parents' positive attitude toward exergames, who preferred exergames to traditional video games [31]. A study with an obese adolescent population qualitatively found that the facilitators to continue using exergames were the feeling of immersion, music, the ease of controlling

exergames, affordable challenges and social cohesion with classmates, while barriers were frustration from technical problems with exergames or challenges were too difficult [28, 32, 33]. The study of Benzing and Schmidt [10] found that the strengths of exergaming were the potential for individualization, adaptivity and specificity, and the capacity to improve motivation and engagement.

Despite the fact that gamification or exergaming is booming in the education world, its effectiveness in increasing motivation has not been proven due to many inconclusive studies [7, 8, 34–36]. For this reason, gaining an in-depth understanding of some virtues of gamification and exergames is necessary to improve learning. We performed a natural experiment in elementary schools using a mixed methods design.

The scientific support for the effectiveness of exergaming and gamification in-class activity is sparse [8, 25], and its effects have been overestimated due to high expectations [8] or the tendency to publish only beneficial results [10]. These findings are interesting *per se* because they raise the following previous research questions with no affirmation bias: "Is a gamified exergaming intervention for school PE applicable and useful or not?", "What are the reasons that make exergaming and gamification more or less applicable and useful?" and "What intervention elements can be improved according to the school community's opinions?".

Hence this research evaluates a gamified exergaming intervention that was designed to improve children's academic performance through psychological variables such as motivation and fun. This paper focuses on understanding the applicability and usefulness of the designed intervention for the ultimate purpose of knowing what didactic intervention elements can be improved according to the school community's opinions. Our operational research objectives were to:

1. To study the applicability of the gamified exergaming intervention using a qualitative method

2. To study the utility of the gamified exergaming intervention using a qualitative method

3. To understand the effects of the intervention separately between gamification and exergames

This qualitative approach allowed us to understand which specific gamification and exergames aspects were more applicable and useful for education, and if these were effects have not been previously considered.

## Study design

The qualitative data herein reported were collected during a natural experiment with a non-randomized controlled design. A qualitative research design was used with field notes, an open-question questionnaire, individual semi-structured interviews and focus group interviews. These data collection techniques are commonly used when the research objective is to profoundly explore a subject about which very little is known [29], and when researchers seek to know participants' reactions to an intervention [37].

Focus groups are especially appropriate when seeking children's contributions for three reasons: they imply a shared intervention experience, which allows to take children's spontaneity; adults have a better chance of understanding what children have to say in their interaction and behavior context; focus groups free children and researchers of data-collection limitations due to literacy/reading levels [38]. The thematic analysis focused on identifying themes related to behavior. The results presented in this study are analytical interpretations of the identified themes [39]. Content analysis was chosen as the methodological orientation.

## Participants

Elementary schools were recruited for this study from May 2017 to September 2017. Of the nine elementary schools invited to participate in it, only one declined because it considered that the intervention incompatible with the curriculum. Criteria Based Selection was used to choose the schools. The researchers chose four schools according to the following criteria: the adequacy of their material (Wi-Fi connectivity, facilities), the diversity of the public/private schools, schools from different cities (Huesca and Zaragoza, Spain), students' ethnic and socio-economic diversity, teaching staff's positive predisposition and accessibility for researchers. Eight teachers took part in the study (75% men, n = 6, 25% women, n = 2; age mean in years = 37.5±6.12; mean of years of teaching experience = 11.5±7.31). The student sample comprised 417 students (53.2% girls, n = 222; 46.8% boys, n = 195) from four elementary schools. Their mean age was 11.1 (SD = 1.7), and 50.4% of the sample studied 5th Grade (aged 10–11 years; 50.9% girls, n = 107; 49.04% boys, n = 103) and 49.6% studied 6th grade (11–12 years; 55.5% girls, n = 115; 44.45% boys, n = 92) in elementary schools.

## Ethics

This research obtained the ethical approval of the Ethical Committee of Clinical Research of Aragon (Spain) on May 24, 2017 (statement n˚.: 10/2017). All the schools invited to participate received a detailed written report of the study. A face-to-face meeting was held with the representatives of each school, when they had the opportunity to ask questions. When a school accepted to collaborate, informational letters and informed nonconsent forms were sent to all the parents/guardians of eligible students. All the children had access to didactic interventions, but only the participants whose parents or guardians agreed to collaborate in the study were included.

## Intervention and materials

A natural experiment with a nonrandomized controlled design was conducted. This design, together with the qualitative approach, allowed good scientific rigor, and has been considered appropriate in similar studies [27, 28, 40]. The control condition (traditional didactic intervention) was designed based on the usual didactic teaching of dance in Spanish PE [41]. Another similar experimental condition was designed to the control condition, except for the presence of an exergame and a gamified design (gamified exergaming intervention). Each intervention lasted 9 hours, which was applied to each school for 4 or 6 weeks during curricular PE classes.

The different number of weeks that an intervention lasted depended on each school's number of holidays. Each school had a different number of PE sessions per week (according to its schedule), but they all had 135 minutes/week. This intervention duration is justified as being the usual way to schedule classes in Spain [41], and is the minimum duration recommended to avoid the bias of "familiarization" with the test [42, 43].

Both traditional didactic and gamified exergaming interventions were applied in the same way by the same male "researcher-teacher". Researcher-teacher was the person who played the role of researcher in this study, but also taught classes in schools. Priority was given to the fact that it was always the same person who taught the teaching units to all the children recruited in the study to, thus, remove the bias of "different teachers" at each school. The control group learnt dance without using exergame and with no gamification strategy.

The gamification system based on the Points-Badges-Leaderboards (PBL) architecture has been extensively applied and studied. However, "achiever" is the only player-student profile that fits that architecture [44], and broader approaches can be made [45]. Accordingly, the Mechanics-Dynamics-Aesthetics (MDA) architecture [46] was used to achieve an inclusive

gamified atmosphere [47] for all the student-player types in the experiment. The MDA architecture was adapted to the education field [48]. Mechanics refers to the system's set of constituent elements, the relation linking them, and the way in which a system can routinely function. Dynamics refers to the way in which mechanics effectively works (adding the player factor and its interaction with mechanics). Esthetics refers to both the perceptions produced by the mechanics in the player-student as it is designed (beauty) and the sensations-emotions experienced by students while playing [48]. All the design elements are specified in Table 1. No negative points of any kind were used. Mechanic elements were designed: a system of positive points, rewards, classifications, levels of difficulty, challenges, achievements, badges, cooperative and competitive teams, virtual avatars and the possibility of personalizing avatars. Self-referential situations in an epic sense [45] of a cooperative group were raised to promote motivation among all the student types, even those who were not competitive [27, 49].

The *Just Dance Now* exergame was used because it can be applied in the facilities of the participating schools, and because it is based on accessible materials for most people in the study context (laptop, screen projector, Internet, and smartphone) [50]. Its use is justified because it has been scientifically studied [51–55] and is extensively used for youth leisure [56]. Playing with a smartphone can be a strategy to obtain a good attitude as young people employ it [57]. The experimental condition design allowed all the students to dance several times during all the sessions, except for the logistical problems reported in the Results.

**Table 1. Gamified didactic design.**

| Mechanics | Dynamics | Aesthetics |
|---|---|---|
| Dance performance point<br>Creativity point<br>Attention point<br>Good behavior point | Reinforcement<br>Cumulability | Pleasure<br>Satisfaction<br>Success<br>Social membership |
| Design a choreography | Self-expression | Pleasure<br>Identity |
| Design a group choreography | Cooperation | Social membership |
| Leaderboard (Fig 3) | Competition<br>Cooperation<br>Progress | Social membership<br>Success |
| Star badge (perfect dance)<br>Green badge (for individual improvement) | Reinforcement<br>Progress<br>Collectability | Satisfaction<br>Fun<br>Pleasure<br>Interest |
| Group green badge<br>Group improvement point<br>Best dancers badges<br>Badge dancers who improved the most<br>Point helping the team | Cooperation collectability<br>Competition<br>Cooperation<br>Collectability | Social membership<br>Identity<br>Success |
| Point of dance plank | Status<br>Competition | Identity<br>Absorption |
| Custom avatar | Self-expression | Customize<br>Identity<br>Beauty |
| Increasingly difficult dance levels | Progress | Fun<br>Satisfaction<br>Interest |
| Music in all classes, from different cultures and periods<br>Choosing dance Level 9 from more than 300 dances | Self-expression | Pleasure<br>Absorption<br>Identity<br>Customize |

In order to gamify learning contents, 10 exergame dances were previously selected from 300 dances of the *Just Dance Now* web platform. The selection criteria were motor difficulty, the dance's cultural variety, and adjusting values to Primary Education: Level 1 "Rasputin"; Level 2 "Crazy Christmas"; Level 3 "Boogie Wonderland"; Level 4 "Aquarius"; Level 5 "Let's groove"; Level 6 "#thatPOWER"; Level 7 "Hungarian Dance no. 5"; Level 8 "I will survive"; Level 9 a dance chosen by the group of students; Level 10 "Jambo Mambo". All the dances are available on the official exergame website (https://justdancenow.com/).

*The ClassDojo* virtual platform was used to make the intervention sessions contextually gamified. Seven badges were associated with seven types of points that are usually given by the teacher to students in class: good behavior, creativity, helping the team, paying attention, motor self-improvement, group improvement, and motor perfection (called "5 stars"). Twelve badges were also designed for the three best dancers at each level, the three best dancers every week, the three students who had improved the most, and the three groups with the most points in general (see Fig 1). Each student's points accumulated in the ClassDojo application and could be consulted by students at any time, even at home. With those points, students could custom their own avatars and obtain a higher score in the subject (see Fig 2).

An *ad hoc* digital game board was designed using the *Microsoft Excel* software (see Fig 3). The students had to enter the score of each dance indicated by the Just Play Now exergame on the gamifier board using a laptop that was available in class. The students entered the obtained scores in the shared computer immediately after dancing each dance during class time. The researcher-teacher arranged a few minutes for this task and helped with management, especially during the first sessions. The board indicated the students' average scores for each dance, and the totals obtained both individually and in groups since the beginning.

## Data collection procedure

The qualitative data herein reported were collected during the natural experiment with a non-randomized control conducted between October 2017 and June 2018 in the four Primary Education schools of Aragon (Spain). Two public elementary schools were selected for this study, and two "semi-private" elementary schools (with partial public funding), both types of schools are very common in Spain (Table 2). All the qualitative measures were collected in the same way: by the same researchers, and with the same procedure. Detailed information on the qualitative data collection can be found in Table 3:

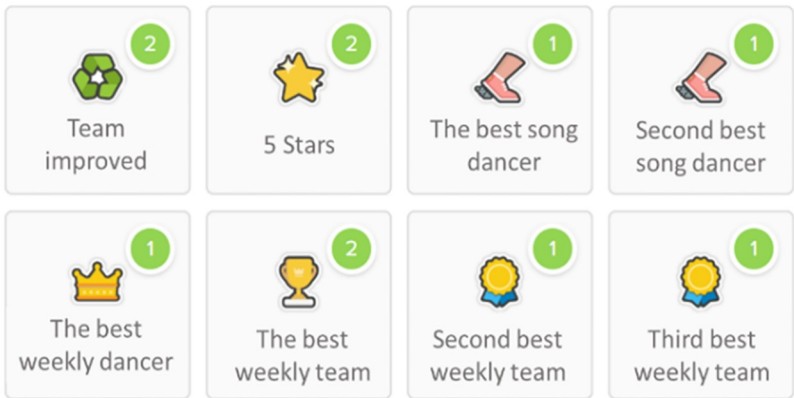

**Fig 1. Screenshot of the individual and comparative badges received by students through ClassDojo.**

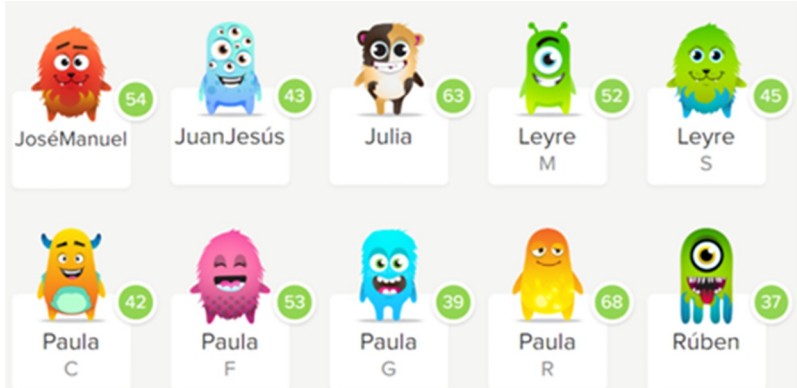

**Fig 2. Screenshot of each student's personalized avatars with the total points of the ClassDojo application.**

1. Field notes: they were collected by the male researcher-teacher during all the intervention sessions. Particular events were recorded, such as accidents, breakdowns, unforeseen events (session changed by another school activity), particular comments made by students, or alternations in the content being worked on.

2. Open-question questionnaire (OQQ): it was administered at the end of each intervention during the last class session to both the control and experimental groups (n = 356). It consisted in responding in writing to four (control group) or six (experimental group) open questions, such as: "*What caused you the most difficulty*?" or "*What do you think about using a video game with which you move in the Physical Education class*?" It took 15 minutes to complete on average. In School 1, a school with a high percentage of students at risk of social exclusion, it was not possible to collect these data because of its students' low literacy skills, and because of their excited emotional state on the last day of class.

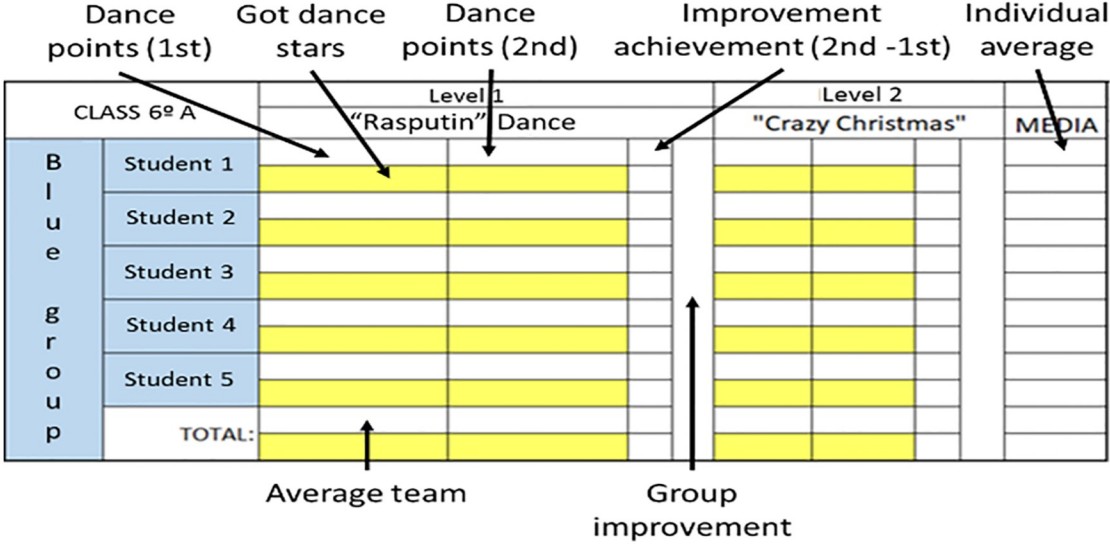

**Fig 3. Partial screen of the gamifier board.**

**Table 2. Information about participant selection.**

| School | Funding | City | Condition | OQQ (n) | | Focus groups (n) | | Interviews (students) (n) | | Interviews (teachers) (n) | |
|---|---|---|---|---|---|---|---|---|---|---|---|
| | | | | boys | girls | boys | girls | boys | girls | boys | girls |
| 1 | Public | Huesca | Control | - | - | - | - | 1 | 1 | 1 | - |
| | | | Experim. | - | - | 2 | 2 | 1 | - | | |
| 2 | Semi-private | Huesca | Control | 19 | 17 | 1 | 3 | - | 1 | 2 | - |
| | | | Experim. | 21 | 23 | 4 | 2 | - | - | | |
| 3 | Public | Huesca | Control | 14 | 18 | 2 | 2 | - | - | 1 | 2 |
| | | | Experim. | 23 | 26 | 3 | 2 | - | - | | |
| 4 | Semi-private | Zaragoza | Control | 46 | 50 | 8 | 8 | - | - | 2 | - |
| | | | Experim. | 46 | 53 | 9 | 8 | - | - | | |

3. Focus group interviews: 14 focus group with four or five students were formed among the four participating schools (n = 56). The student's choice was made by the male researcher-teacher according to his/her experiences throughout the intervention. The selection criteria were: attitude shown by students (very positive and very negative), student's gender, student's communicative ability and student's agreement to be interviewed. All the interviews were filmed and audio-recorded, always transcribed by the same researcher, and re-imputted into different files to ensure data protection (Transcripts). The interview comprised open questions, such as "*Do you think your dance skills have improved*?". Debate was encouraged among students. This part 20 minutes on average.

4. Individual semi-structured interviews: at the end of each intervention (both control and experimental), a personal interview was held by the same two researchers with some students (n = 4) and all the participating teachers (n = 8). Only certain students were interviewed if they were considered able to provide in-depth information. The choice of students was made by the researcher-teacher according to his/her experiences throughout the intervention, and when it was observed that the student's profile could be very interesting: two students were chosen for increasingly showing special interest in classes during the didactic unit, and two students for quite the opposite.

The interview consisted in the same open questions for students and the focus groups, with other open questions for teachers, such as "*What advantages and disadvantages do you see in using a dance video game as an educational resource in school PE*?" (Transcripts). Each teacher knew both interventions (traditional and gamified with exergames), as did an assistant professor to the male researcher-teacher who performed all the interventions, which meant they knew both didactic methods. This part took 25 minutes on average.

This data collection strategy allowed mixed triangulation: triangulation of observers (teachers and students), triangulation of researchers (researcher-teacher and researcher 2) and triangulation of instruments (interviews, discussion groups and OQQ).

**Table 3. Information about the qualitative measures.**

| Instrument | When? | Who? | To whom? | How long? | How many people at a time? |
|---|---|---|---|---|---|
| 1. Field notes | during the intervention | researcher-teacher | any aspect | 1–3 minutes each annotation | - |
| 2. OQQ | last session | researcher-teacher | students | 15 minutes | 12–25 |
| 3. Focus groups | after the intervention | researchers 1 and 2 | students | 20 minutes | 4–5 |
| 4. Interviews | after the intervention | researchers 1 and 2 | students and teachers | 25 minutes | 1 |

## Analyses

The information collected by OOQs were transcribed and categorized emergently using the Spss (version 21.0, https://www.ibm.com/es-es/analytics/spss-statistics-software) software by the same encoder. These data were quantitatively processed in order to triangulate them with the information collected from the interviews and group discussions. The analyses consisted in descriptive contingency tables by comparing the control and experimental groups (in the common questions), and by focusing only on the most frequent or interesting open answers.

A thematic analysis was conducted [58], which was both deductive (to study the previously raised problems and elements) and inductive (allowed new issues to emerge). The original classification tree was built based on the previously considered concept. The first analysis layer derived from the first and second study objectives; that is, the applicability and utility dimensions, which are basic study dimensions. The second analysis layer derived from the third objective by differentiating "exergame", "gamification" and "condition as a whole". Operationally, "applicability" has been defined as the ability of the gamified exergaming intervention to be carried out at school easily [facilitators], with minimum obstacles [barriers], with the best predisposition of teachers [attitude, expectation of use and continuous learning] and students [attitude, affinity], enduring in time [continuity] and to develop the educational curriculum. "Utility" has been defined as the ability of the gamified exergaming intervention to produce concrete and direct effects of psycho-educational interest, both as a whole [as a didactic design] and through some of its characteristics [gamification, exergame, dance, etc.]. Two similar, yet different, classification trees were built for teachers and students.

When the original classification tree was built, two researchers used it to jointly analyze a student's interview, a teacher's interview and a focus group, and added emerging categories. All the operational definitions used in each tree category can be seen in the S1 Material (Classification Trees). The Nvivo software (version 11, https://www.qsrinternational.com/nvivo/home) was used to analyze all the content from interviews and focus groups. These emerging categories were the fourth analysis layer based on participants' responses. This way in which the final classification tree was achieved allowed a deductive and inductive analysis [58]. Next the reliability index between the two researchers ($\kappa$ = .94) and intrapersonal reliability ($\kappa$ = .94) was calculated. This last researcher was the encoder who analyzed all the content of the interviews and discussion groups. Children and teachers was anonymized. Excerpts from the interviews and focus groups for each theme are included in the S2 Material (Transcripts), along with coding sheets with more details about the qualitative analysis (Classification Trees). A COREQ checklist with further details of the qualitative design is also included in the S1 Checklist.

# Results and discussion

## Quantitative indicators

The OQQ allowed us to find that the greatest difficulty expressed by students (control = C, Experimental = E) lay in working in groups (C = 38.4%, E = 44.3%), complexity of dances (C = 44.5%, E = 19.3%) and body expression activities (C = 5.5%, E = 17.2%). What they least liked about the intervention were body expression activities (C = 18.9%, E = 36.5%), adapting to the song of the final choreography (C = 42.7%, E = 13%) and making group decisions (C = 12.2%, E = 14.1%). Students would have liked more freedom to choose the groups and songs (C = 46.3%, E = 24.5%), a more current repertoire of songs (C = 15.9%, E = 8.9%) and more practice time in the exergame (E = 22.4%). The most frequently shared feelings were enjoyment (C = 32.3%, E = 55.2%), embarrassment (C = 20.7%, E = 16.7%) and positivity (C = 14%, E = 8.3%).

Descriptive analyses were previously performed with the focus groups and interviews (see Tables 4 and 5). This information was used as an indicator to direct the thematic analysis. More specific descriptive analyses can be found in the appendix (Classification trees).

The qualitative and the thematic analyses of this study are presented below. The results are presented through textual extracts taken from different interviews and discussion groups. The coding used to identify each extract was based on four digits: the first number refers to the specific interview or discussion group (numbered from 1 to 26); the first letter indicates gender ("B" for boy, "G" for girl); the second number indicates the paragraph number in each transcribed document; the second letter indicates the category of participants ("s" for student, "t" for teacher).

## Applicability

**Logistics.** The gamified exergaming intervention was used in four schools with very different characteristics (students, teachers, materials, installations, etc.). This means that despite making logistical and organizational decisions and changes, the same learning sessions were held at all the schools, educational objectives were met and an equivalent evaluation was made. Of the main intervention facilitators, the realistic design for the school context as an open changing setting stood out, as did its adaptability to be applied to differing contexts without it losing its didactic essence. This result was acquired thanks to the 9-month researcher-teacher experience, which was triangulated with the views perceived by the teachers participating in the four schools: "*We have a block and some objectives, and the exergame helped to work by facilitating a context, because you had quite a specific context and a place where you knew what you were going to do*" (1B5-t).

**Table 4. Descriptive analysis of the focus groups and interviews with students.**

| Classification tree | C Group | | E Group | |
|---|---|---|---|---|
| | N Cod. | % Cod. | N Cod. | % Cod. |
| 1. Applicability | 29 | 18.01 | 34 | 20.00 |
| 1.1. General design | 29 | 18.01 | 34 | 20.00 |
| 1.1.1. Changes | 11 | 6.83 | 16 | 9.41 |
| 1.1.2. Strengths | 17 | 10.56 | 18 | 10.59 |
| 2. Utility | 119 | 73.91 | 126 | 74.12 |
| 2.1. General design | 119 | 73.91 | 115 | 67.65 |
| 2.1.1. Enjoyment | 22 | 13.66 | 21 | 12.35 |
| 2.1.2. Learning | 37 | 22.98 | 25 | 14.71 |
| 2.1.3. Academic performance | 27 | 16.77 | 30 | 17.65 |
| 2.1.4. Perceived physical effort | 10 | 6.21 | 9 | 5.29 |
| 2.1.5. Usage expectations | 1 | 0.62 | 11 | 6.47 |
| 2.1.6. Promoting PE | 10 | 6.21 | 10 | 5.88 |
| 2.1.7. Resolving conflicts | 12 | 7.45 | 9 | 5.29 |
| 2.2. Gamification | 0 | 0.00 | 11 | 6.47 |
| 2.2.1. Advantages | 0 | 0.00 | 9 | 5.29 |
| 2.2.2. Disadvantages | 0 | 0.00 | 2 | 1.18 |
| 3. Miscellany | 14 | 8.70 | 7 | 4.12 |
| 3.1. Researcher contributions | 12 | 7.45 | 7 | 4.12 |
| 3.2. Irrelevant information for this study | 2 | 1.24 | 3 | 1.76 |

"N Cod." = number of times that the variable was encoded. "% Cod." = percentage of the total encoded references.

**Table 5. Descriptive analysis of the interviews with teachers.**

| Classification tree | N Cod. | % Cod. |
|---|---|---|
| 1. Applicability | 94 | 39.50 |
| 1.1. Exergaming | 50 | 21.01 |
| 1.1.1. Facilitators | 11 | 4.62 |
| 1.1.2. Barriers | 39 | 16.39 |
| 1.1.2.1. Logistics | 8 | 3.36 |
| 1.1.2.2. Students | 0 | 0.00 |
| 1.1.2.3. Technical problems | 15 | 6.30 |
| 1.1.2.4. Specific teacher training | 7 | 2.94 |
| 1.1.2.5. Critical education | 6 | 2.52 |
| 1.1.2.6. Teaching dependence on technology | 3 | 1.26 |
| 1.1.3. Usage Expectations | 0 | 0.00 |
| 1.2. Gamification | 1 | 0.42 |
| 1.3. General didactic design | 43 | 18.07 |
| 1.3.1. Strengths | 21 | 8.82 |
| 1.3.2. Changes | 22 | 9.24 |
| 2. Utility | 113 | 47.48 |
| 2.1. Exergaming | 58 | 24.37 |
| 2.1.1. Benefits | 54 | 22.69 |
| 2.1.1.1. Learning | 13 | 5.46 |
| 2.1.1.2. Health | 2 | 0.84 |
| 2.1.1.3. Performing alternative leisure | 7 | 2.94 |
| 2.1.1.4. Promoting PE | 14 | 5.88 |
| 2.1.1.5. Motivation toward learning | 18 | 7.56 |
| 2.1.2. Prejudices | 4 | 1.68 |
| 2.2. Gamification | 35 | 14.71 |
| 2.2.1. Benefits as a method | 29 | 12.18 |
| 2.2.1.1. Motivation toward learning | 17 | 7.14 |
| 2.2.1.2. Promoting PE | 2 | 0.84 |
| 2.2.1.3. Promoting cooperation | 4 | 1.68 |
| 2.2.1.4. Facilitating teacher evaluation | 4 | 1.68 |
| 2.2.1.5. Achieving learning | 2 | 0.84 |
| 2.2.2. Prejudices | 6 | 2.52 |
| 2.2.2.1. Too much competitiveness | 4 | 1.68 |
| 2.2.2.2. Points dependence | 2 | 0.84 |
| 2.3. Utility of the general didactic design | 20 | 8.40 |
| 2.3.1. For teaching-learning processes | 15 | 6.30 |
| 2.3.1.1. Promoting curricular values in students | 5 | 2.10 |
| 2.3.1.2. Resolving conflicts | 3 | 1.26 |
| 2.3.1.3. Overcoming problems | 7 | 2.94 |
| 2.3.2. For health | 0 | 0.00 |
| 2.3.3. Performing alternative leisure | 2 | 0.84 |
| 2.3.4. Promoting PE | 0 | 0.00 |
| 2.3.5. For motivation | 3 | 1.26 |
| 3. Miscellany | 31 | 13.03 |
| 3.1. Researcher contributions | 24 | 10.08 |
| 3.2. Irrelevant information for this study | 7 | 2.94 |

"N Cod." = number of times that the variable was encoded. "% Cod." = percentage of the total encoded references.

The main barrier to apply the intervention was the logistics of materials and spaces: "*I think that its main problem is that it involves a number of minimum technical elements*" (1B11-t). One of the problems that most arises is when space is hard to find and must be shared by many teachers: "*I´m not sure if it is our problem or the school's problem but perhaps before beginning the unit, who is in charge of managing spaces, this must be made quite clear.* [. . .] *Because I found that this was a problem*" (07G17-t). This female teacher also stressed the problem of spaces not being coordinated, which was specifically due to her school (School 3) that was smaller and more modest, although the same problem arose sporadically at School 4. The research experience in this study, along with field notes, indicates that, quite often, it was never really a coordination problem, but was due to a real problem caused by lack of space. One reason for this is because PE is normally practiced in places that differ from where other subjects are taught (pavilion, playground, swimming pool, the natural environment). Nonetheless for this intervention, classes had to be given in each school's Assembly Room (to set up a projector and the Wi-fi network). These places tend to be frequently requested by more teachers to perform different activities in distinct courses.

The logistics problem of spaces and installations cannot be completely solved by applying more coordination at school given its daily open dynamic character. At any time, spontaneous activities may arise at school that change its planning: "*some events at schools are unavoidable, like the International Day of Peace* [. . .] *then class starts 30 minutes later.* [. . .] *if possible,* [*it is necessary*] *to control such events.* [. . .] *by the required logistics, organization, especially for materials and spaces. As for the rest,* [. . .] *it all worked well for us and, fortunately, the means you have when you have brought mobiles and the rest; the spaces we have used; projectors; Wi-fi. . .*" (12B23-t). Contextual problems of this kind are mentioned in similar studies [28]. The field notes more frequently indicated at School 4 (the school with more people) that the bookings previously made for this intervention were spontaneously amended by other activities that were considered more preferential by the school. This is because the Assembly Room is used for those events that are particularly important for the school (meeting with parents, presentations, etc.). It can be stated that each school's Assembly Room allowed the intervention classes to be carried out, but it is necessary to seek alternatives for the project to be consistent over time. Moreover at most schools in Spain, this intervention is applicable only by coordinating and managing spaces and materials well. Equipping the usual places where PE is taught with digital technology (projectors, Wi-fi) would solve the high demands to coordinate and plan spaces: "*Now we have this, and we have performed it in the Assembly Room, we can change and try it somewhere else. Or we can move it. But then again, if you don't try doing that, you can't see the failures or benefits*" (12B17-t).

A similar study detected general lack due to the chosen exergame barely being used [28] as it was destined for recess time or for out-of-school activities. However in the present study, the official curricular time was used and activities took place in the Assembly Room.

The school's dynamic nature strongly influences an intervention like that herein applied because it requires being set up and adapted before holding classes that involve using another type of school contents: "*when it is necessary to prepare things for PE, I always arrive much earlier, and I try to leave the gym ready before a class precisely to play with that motivation.* [. . .] *When they see everything set up, it´s a plus. So I would go further as to whether you can calibrate* [spaces] *in centres, which I know is difficult, or even impossible, in many aspects*" (12B23-t). This proposal to improve the logistics of materials was hard to plan well in advance and in much detail; more often than not, previously adapting the classroom was not possible because another teacher was giving a class in another course. However, as the cited reflections suggest, having installations previously prepared at the start of class, whenever possible, is believed necessary to create more student motivation and to optimize time. Having the necessary time

beforehand to adapt to this gamified exergame session would be a limit to apply it, which is not unusual with other PE contents.

Another logistic problem indicated by teachers was related to materials (smartphones): "*the limitation of having to rely on mobile phones; the limitation of the material, I believe that we at schools are not ready for all this, as we do not even have proper networks*" (6B5-t). Despite considering the intervention to be applicable in this way (teachers are used to not having material for all their students), they mentioned the need for good group organization capacity: "*It can be done in groups with everyone dancing, and this week we have this song; then on such and such a day, you haven't got exergame, so you have the time to practice and it's your turn first next week* [with exergame] *and you have to make so much effort. I mean, there are different ways of using it, but it involves that planning*" (01B11-t). The intervention increases its applicability if help is sought to plan it with students: "*Then you need previous preparation work, knowing well what we want to do, how we want to do it, students who are more used to dealing with exergame, video games. . ., students can perhaps help you during this session*" (06B16-t). Apart from better organization, three teachers from different schools considered a feasible solution to collect old mobile phones for schools, exactly as done before with another recycled material for PE: "*When I suggest obtaining second-hand* [smartphones], *I mean they can be borrowed from people, or donated.* [. . .] *It would just be a matter of organizing a small campaign with teachers, parents . . .*" (11B10-t).

This alternative to obtaining the material would be a minor advantage over other studies that have used exergames and required materials such as dance mats [28, 59], which are more expensive and more difficult to manage. However, smartphones have presented similar problems to dance mats, such as run down batteries, outdated models or technological failures [28, 60].

In this sense, the public or semi-private character of the schools in this study did not seem to affect the logistic variables of the spaces and materials differently from the teaching staff's perspective. That is, some comments were made about difficulty in the variables of space and materials by teachers from different schools. In any case, it may be convenient to consider such aspects when determining the applicability of this type of resources by taking into account that applying an exergaming program may depend on the center's available resources [61], and some studies have already considered the type educational center to be a covariable [62]. Thus, this main logistics barrier coincides which what other studies have indicated [28], which involves not only the applicability of the intervention, but is also one of the limitations of such research because exergames are not used so much in scientific-didactic interventions as expected.

**Novelty and tradition.** The first response with which the male researcher-teacher came across when considering the experiment in all the participating schools was teachers' skepticism, along with their doubts about the intervention's applicability: "*It is true that when you presented the project to us, at first I thought it would be impossible because I did not understand it. I mean, I did not understand its dynamics, how it could be organized*" (6B22-t). When the process ended however, teachers' overall attitude was positive: "*I quite liked it because we began with simple things that were related. I mean nothing isolated was included, nothing that did not work toward meeting the ultimate objective, rather everything worked toward that objective*" (17B3-t); "*My first impressions were positive because I noticed that students were really motivated*" (18B3-t).

Teachers considered the intervention applicable as they thought it was compatible with conventional didactic methods: "*I believe that virtue lies in mixing things. Not just one thing, not all exergames. . .. precisely what we have done. It might be better doing half a unit as always and then adding strokes of exergames, why not? Or alternating*" (12B17-t). The compatible nature of exergames and gamification came over by it presenting PA and allowing students'

curricular knowledge to develop: "*I think that any advantages that we embrace for students' knowledge are positive.* [. . .] *So, is this any different? Yes. Is it considered a form of physical activity? Yes.* [. . .] *I´d propose anything that matches our contents and objectives*" (18B17-t). This coincides with another qualitative study that detects that PA is a benefit and a reason for teachers to include exergames [28].

Despite the initial doubts raised, teachers welcomed the novelty and considered the intervention to be applicable because it is compatible with the school curriculum, and also with the PE area for both its didactic methods and contents: "[exergame and gamification] *are technical elements to develop the curriculum*" (15B-t). These considerations were formed throughout the intervention, and were positive halfway through and at the end of the process, when teachers could see and understand the class dynamics. Students' changed attitude was also a determining factor: "*Especially with what I see when having to work, which is motivating. When using an element, but not just having a phone in my hand, but also what is visual. Then there is the feedback given when scoring: you finish and you have points. Then you think, well that's it, and you know if you have done it well, or badly. Then you think I´m going to dance again and see if I can get a better score . . . I think that this feedback is very important too*" (1B7-t).

Novelty is one of the reasons mentioned in a discussion group of the E group to show a positive attitude: "*we like it because it is something we have never done before. We have never tried doing things like this, dancing in PE*" (23B26-s); "*And with video games*" (23G27-s); "*Then they scored us, and I really liked it*" (23B28-s). Three different novel elements stemmed from this conversation: dancing as content, exergame and gamification. However, the C group also referred to a novel aspect: "*I personally liked it a lot because I had never done anything like this. We have always done sport games in playtime, and this year has been the first time we have used dance. It has been great fun*" (20B6-s). Salsa was also included as novel content in the C group so that it matched the E group: "*I really liked salsa because I was having fun with my friends and because it is a dance we´d never practiced before*" (20B5-s).

**Expectations of its use for teaching.** The attitude shown toward the possible future application of such these interventions was not always the same for all the teachers who participated in this study. A positive attitude was shown toward the study intervention as it was shown to be feasible: "*I have seen that linking technology and education, PE in this case, is possible*" (18B7-t); "*Yes,* [I would apply this intervention again]. *And in exactly the same way as we have already done* [. . .] *Moreover given the experience, which has been positive, we have the space and resources to use it, so there would be no problem*" (12B17-t). Likewise, a generalized positive attitude was shown toward gamification and exergame as general educational resources: "*Oh yes,* [I would use exergames as a resource]" (17B31-t); "*considering gamification and using technologies to develop units* [dance lessons] *should not be a problem*" (01B15-t). Some teachers had good expectations of its use for teaching, even depending on the required logistics: "*If I had the materials, yes* [I would use exergames and gamification again]. *I really enjoyed it, honestly*" (18B19-t).

However, a more skeptical attitude was shown by other teachers, who found it hard to acknowledge their priorities, perhaps due to the social desirability effect [63]. Given this suspected bias, the interviewers asked the same questions, but they were expressed differently and they spent longer on this theme. A previous condition for teachers to apply the intervention in the next academic courses was for them to have sufficient training and knowledge: "*I will try it, but my knowledge of technology is limited. So I would use it if I was sure about the content I´m teaching*" (6B15-t). Some teachers indicated not having such training: "*You see in my case, I don't control the themes, it's not a content I handle. I have never played and I don't know how to play. So my knowledge is very poor*" (6B16-t). This lack of specific competence has been found in similar studies [61]. Therefore, this intervention would be more applicable if teachers had

more specific training in exergames and gamification. However, the required training was not placed as a priority because teachers carry a heavy training load in other new trends: "*I don't think that exergames is a priority line for the PE department, simply for priority reasons, and because PE can be done well, and it can be made motivating and creative without having to work with exergames. I don't believe it is an immediate priority*" (12B27-t). Exergame was perceived by some teachers as being interesting for PE, but not essential, which falls in line with [28] as it is not the only resource that allows certain objectives to be fulfilled, like motivation. Another male teacher stated something similar: "*now there are many options open as regards new methodologies. [. . .] There are many things available. Well, of course, it can be included, but it won't be easy. It's not a priority right now*" (11B24-t). This might lead us to think that teacher expectations were low because, despite lacking training, they felt it was not a priority to train in exergames and gamification. This skepticism coincided with the views that the teachers in [27] indicated.

**Digital technology in class.** When the intervention ended, a positive attitude was shown by some teachers for using the video game in PE, if it was like exergame, because it was associated with PA and health: "*I believe that video games are not well considered socially speaking, [. . .] so, [it is necessary] to give a point of view, which also favours healthy life styles. We are talking about dancing, performing physical activity, sport. [. . .] It is like combining two things they like, changing the negative parameter, let's say the game. Because, at the end of the day, they use it to practice physical activity. Which is exactly the opposite to what other video games achieve*" (12B5-t). Therefore, some teachers did not perceive exergame as being linked to PE, but as another instrument available for educational interests: "*for me exergame is an instrument you can use to fulfill the objectives you set out, your educational purposes*" (1B5-s). One male teacher even postulated an artificial theoretical barrier between video games and sport: "*Perhaps we adults spot a difference between sport and video games. And if we did not find this difference, they might play something else*" (17B21-t).

Other teachers, however, mentioned the technological nature of exergame as something negative. In the first case because (digital) technology means losing something essential of PE, and perhaps interactions could occur in the "natural" environment: "*I prefer nature myself, playing with sticks, stones, etc., which we are missing out on. With the few hours we spend on PE, I think we should focus on forgetting to use mobile phones, technologies, and we should act as a group that only interacts*" (7G7-t). In the second case, students' early age was mentioned as the main motivation to oppose including the use of mobile phones as an educational resource: "*I don't agree with them having mobile phones and being able to play with their phones, not in year 5, year 6, nor even in Secondary Education, especially in years 1 and 2*" (11B8-t). Another male teacher believed that technology could lead to dependency and teachers not being able to control: "*Disadvantages, apart from practical ones, include us depending on technology that is, therefore, a resource we cannot control. When you are teaching, you manage your class, you manage times. As soon as you depend on networks, spaces and a technology, times differ*" (6B10-t). These teaching considerations were preceded by an initial reluctance shown by the schools' different Management Teams to introduce smartphones into class as an educational tool. They finally accepted to participate when they explained that their use was designed for purely educational purposes and in a controlled manner. This attitude, which was one of the main barriers to apply the intervention, also comes over in several studies conducted in other countries [27, 28, 64, 65].

The various conceptions of exergames have implications for how teachers use them, which more determine the use of exergame than the nature of exergame [61]. In this study the instrumental conception dominated, which did not coincide with the teachers who participated in another qualitative study, in which exergame was not considered by them as being essential in PE; that is, in what is sport [61].

Students generally showed a very positive attitude toward exergame. However, they indicated that they found systematic technological failures frustrating: "*What I didn't like was that my phone switched off when I held it in my hand*" (5B48-s), which has been previously reported [28, 32], perhaps because it leads to boredom [28]. At all four schools, the Wi-fi signal was interrupted, which did not allow the exergame activity to continue and led to students feeling frustrated. This is a major barrier for the *Just Dance Now* exergame, as is also indicated in other studies with different exergames, like *FitQuest* [27] or *XaviX Bowling* [33]. Although no mention was made about a smartphone being a strange element to dance with, comments were made about the inconvenience of having to hold them in a certain way to avoid them switching off: "*My arm ached holding the phone*" (5G45-s). This intervention was done with children aged 10–11 years and with smartphones that were the standard size for adults. One of the negative aspects mentioned was that technology was a limitation as regards the number of available devices; despite the theme being centered as a change to share and manage materials in groups, one female student mentioned that "*what I did not like much was that while others were dancing with mobile phones, some others had to dance with no phone. This was no good because you didn't score points*" (10B3-s). To overcome this limitation, an equal distribution system was created that would not affect either the individual or the group in terms of absolute points, and the order was set by students themselves voluntarily, who were awarded another type of points to those they shared and gave. Notwithstanding lack of material may have led to some frustration.

The male researcher-teacher indicated other technology-related technical failures in the field notes, and was able to solve some at the time, but had to seek alternatives with others for exergame. The most frequent failures were: the Wi-fi signal not reaching the classroom, the smartphone not recognizing the Wi-fi network (and not collecting dancing points), and the smartphone not running exergame.

Another failure associated with technology was the *Class Dojo* application. Students were delighted with the chance to see their own results for the application at any time: "*I liked the idea that an application is used in which we can see for ourselves if we want to see our points, rather than having to ask about them*" (23B134-s). However, the field notes reflected that the application was not used much outside the classroom. Many students indicated in class that they had unsuccessfully attempted to consult their report: "*I tried, but couldn't see it*" (23G135). This was because parents are required to participate (they have to consent their children using the application) and many participants did not obtain this beforehand. Technical problems also arose with the application because the procedure was not known to authorize this permission to their children. An alternative was to show the personal reports to those students who requested them at the end of each class.

**Students' affinity.** Elementary school children's possible affinity to video games and gamification today could make an intervention like that studied herein more applicable. Some teachers showed a positive attitude to technology as their students are used to such habits: "*I liked it, especially children's reactions when working in PE with a means like technology. You can really see that they live in the technology era, and their interest and motivation quickly grow when using this small . . . [technological component]*" (07G3-t). Another male teacher also considered exergame to be motivating thanks to students' affinity to technology: "*with benefits firstly in motivation because our students get their hooks into anything technological. They're used to working with new technologies*" (6B9-t). This affinity has already been recognised in other qualitative studies, and by parents [29, 31] and teachers [27, 28].

Students generally reflected on liking the intervention: "*I generally liked everything*" (5B5-s) and mentioned their different reasons, like it being fun, presence of music, which coincides with that reported in [32], or showing an interest in something new. This positive attitude is

also found quantitatively in students of the same age [25, 30]. Most student contributions to improve the intervention's affinity to them, and to make it more applicable, are summarized as: choosing more modern songs (in both the C and E groups) [61], making fewer corporal expressions, which are uncomfortable (more dancing in the C group, or more exergame in the E group) and being able to select more songs. These student requests are coherent with the Self-Determination Theory [66] and coincide with what another similar study indicated [67], which proposed generating activities desired by students to enhance intrinsic motivation.

In short, the gamified intervention with exergame came over as being partly applicable. The facilitators were the realism of the didactic design and its adaptability to different educational contexts. The obstacles that hindered its applicability were not minor, and stemmed mainly from lack of materials (smartphones), the need for spaces in conditions that are not often requested (room with Wi-fi and a projector), and a powerful Wi-fi network signal, which means that this intervention strongly depends on material. The solution proposals included better coordinating spaces in schools and acquiring smartphones by voluntary collections. The teaching predisposition toward the intervention was not conclusive as a positive attitude was shown toward it, but future expectations of training in the subject were low, and the expectation of future use was questioned by one male participant: negative use expectation given its technological nature or not being considered essential; a positive expectation of its use was denoted by students' affinity to it, its novelty and its psychological effects. Initially, the Schools' Management Teams were quite reluctant about including mobile phones. The intervention appeared highly applicable given students' positive attitudes and the good affinity they showed to its technological and video game nature. The intervention was feasible with time and strongly dependent due to previous aspects, especially in terms of logistics and teaching expectations. In curricular terms [68], the intervention proved compatible as regards both contents and usual PE methods, and came over as being even more efficient than the C intervention for variables like being fun, motivation, autonomy or lack of corporal inhibition.

## Utility

**Being fun and motivation.**   Being fun was the feeling that the students most frequently mentioned about the intervention in general. No many references were made to negative aspects. The fact that the students thought the intervention was fun was associated with exergame's game character [25, 68], the presence of music, or certain gamified elements like scoring: "*I thought it was great and really fun because you play as if it was a video game at school, moving about, it was cool*" (24B3-s). This finding coincides with similar studies that had qualitatively found that fun was one of the main strong points of exergame [27, 29, 31, 61]. It also coincides with what was collected by the OQQs and by the team's field experience. The most frequent descriptions made of this intervention were "fun" and "cool". Most teachers mentioned that motivation was the main benefit of the intervention, and it was even mentioned by those teachers who were negatively predisposed to video games: "*My view about technologies and video games is very radical. So in line with my view, its benefit is that you have motivation and interest*" (7G5-t). This finding coincides with that reported by the teachers in study [27], who explained the fun of using the exergame PBL system.

**Shame.**   The main difficulty for both groups (C and E) lay in the corporal expression activities given their lack of previous experience and them not being used to them: "*I felt a bit ashamed, even though no-one was watching. I felt ashamed just thinking about it. I didn't feel at ease*" (14G14-s); "*I found it a bit hard because when it was your turn to perform corporal expression, you think «I don't know what to do»*" (15B3-s). One female student in the E group indicated feeling less shame with *Just Dance Now*: "*Sometimes, I mean before I felt ashamed to*

*dance with people around me. But when we worked with Just Dance Now, I no longer felt ashamed because I danced casually*" (5G22-s). One male teacher indicated that "*They find this lack of inhibition part hard. Then they later have to teach others. That is where the barrier appears, which they find a bit hard to overcome. With exergame, this shame barrier is quickly overcome*" (11B6-t). The C group improved throughout the intervention. Therefore, the potential of exergame to overcome shame is not altogether clear: "*I, [have greatly ovecome] feeling ashamed. I mean, before I focused more on what people would think, but not anymore. I don't really mind what they think now*" (14G20-s). One male teacher stated that a material advantage of exergame for lack of inhibition was the presence of a screen, not focusing on other classmates, and perceived competence increased: "*they no longer feel so ashamed because they are not being watched while their classmates dance, rather everyone is watching the screen and they feel more capable because they are immediately recognized by scoring points*" (17B5-t). These remarks about shame coincide with what the OQQs found, where shame was the second most experienced feeling. Likewise, these remarks coincide with having more difficulty expressing corporal expression, which was one of the activities that the students least liked. Exergame came over as an alternative for those youths who felt ashamed in not only corporal expression, but also when practicing PEx in public [28, 67].

**Learning.** The intervention was considered to benefit student learning in three ways. The first was direct motor learning: "*Just Dance Now helped us to learn new dance steps*" (16G32-s); "*I have worked my coordination very well by dancing in parallel to the screen*" (11B8-t). The second was significant learning; that is, conscious: "*It helps me see how I had improved*" (16G67-s). Thirdly, it improved the motor competence perceived in the subject that derived from perceived learning: "*I think my dancing has improved, and I have learned new steps. I have improved a bit in PE in general because we move differently, and we can move parts of your body in another way*" (3B7-s). Such learning was also a benefit reported in other qualitative studies [27–29, 67], and other studies with different methodology [62, 67, 68].

Exergame was not didactically included only as a repetitive learning activity, but was framed within a creative process to build a choreography, which was perceived by the students. Despite both groups stating that they had learned, only the E group acknowledged that the preparation activities of an invented choreography acted as a source of inspiration: "*in Just Dance Now, if you don't know how to dance and you don't know the movements to create a song [. . .] then you have ideas about what you can do next*" (24G6-s); "*I think that dancing was fun because we got ideas from Just Dance Now*" (15G6-s). The male researcher-teacher was able to see how the E group was generally more autonomous in creating the new choreography, perhaps because it obtained more sources of inspiration with exergame. Different teachers made this observation about autonomy: "*I think the year-6 students* [E group] *wanted more and were more autonomous and you did not need to tell them off or be on top of them as much by saying «this must be done, that must be done». The year-5 students* [C group] *did not find it so easy*" (17B25-t); "*it is true that no-one sat down with Just Dance Now. I don't like admitting this* [. . .], *but when we danced, some sat down and I had to make some of them get up off the floor and ask «what are you doing?»*" (18B33-t). It seems that exergame enabled students to better focus on the task by allowing them more autonomy as they required less continuous teaching intervention. This facet can become an organizational advantage because it allows teachers to center on other didactic aspects in class.

**Promoting physical exercise.** The students related exergame with PEx and health: "*apart from enjoying dancing a lot, and a lot of people like it* [. . .], *you are also practicing sport for your health*" (24B4-s). Many students indicated that they liked exergame and they had downloaded it at home: "*Yes, [I would play exergame in my free time]. I played it the second week after we did in class. I have started playing Just Dance Now at home. I didn't know it before. I didn't*

*know this application and I like it. At home, I sometimes play with my cousin, and alone on other occasions*" (3B15-s). However when given the chance to choose, it would seem that those students who liked and were used to PEx traditionally preferred to continue that way "*I think that both things are good. If I had to choose, I prefer to go outside*" (23B142-s); "*When I feel like doing exercise, I always dance. I have no video games, but if I did, I´d prefer to dance with no game*" (25G64-s). For those who initially came over as leading sedentary lifestyles, exergame was not such a great discovery to help them change their habits: "*this I did at school, but in my free time, I want to do what I feel like, and I´d actually prefer to keep still*" (23G147-s). In particular, those students more used to the world of video games appeared to be more receptive to this intervention: "*I like playing Just Dance Now more because it's like a video game. I´m used to playing video games, and I more or less know how it works*" (3B16-s). These remarks coincide with [28, 69], which report how exergame acts as a strategy to promote PEx only in those students used to playing video games. They are also coherent with another study, in which the students of the same age were willing to use exergame, while adolescent students preferred outdoor activities [29].

Other studies have reported failure to promote PEx through exergames because the participants consider they are boring [70]. This reason did not appear in our study. The ability of exergame and gamification to perform PEx is not yet clear, and more studies must be done.

Some teachers in a qualitative study [28] conceived exergames as a useful resource, especially for those students who had sedentary lifestyles. Similarly, teachers believed that exergame was a good alternative, but only as a passive digital entertainment formula, and not for traditional PEx: "*If there was some way of performing physical activity in the natural environment, with other classmates, don't let technology be involved, then I'd firmly go for that option. However for those students who spend many hours at home, it's an excellent way for them to spend their free time*" (7B14-t). Another male teacher from a different school thought along the same lines: "*I think that it is much better to perform physical activity outdoors, but I believe that your work is an alternative for those kids who, for whatever reason, do not spend much time outdoors*" (18B9-t); "*I think that we have plenty of very interesting resources apart from* [digital] *technology to make movements and to perform physical activity*" (18B29-t). Exergame allowed people to recognize its benefits, which make it a valid option to perform PEx, but it was less preferred for PEx done traditionally by sports. This view voiced by teachers coincided with what a similar study found [28], where one male teacher believed that "*no computer games would ever take the place of sport* [. . .], *but at least it is some form of physical activity*".

## Differentiating gamification and exergame

The most frequently reported student valuations of the gamified system were a general positive assessment (28.6%), its motivating character (28.1%) and it being informative (13%), or its ability to individualize feedback (12.5%). Negative comments referred to a feeling of indifference (6.3%), stress (4.7%), frustration (2.6%) or nonsense (2.1%). The most frequent opinions about introducing exergame into PE classes were associated with fun (38.5%), nonspecific positive assessment (21.9%), a wide range of songs (18.2%) and usefulness for learning (15.6%).

Although not all the students showed the same attitudes to and preferences for video games, it can be generally considered that the main source of motivation came from the gamified atmosphere [62]. The points system as a gamifying strategy allowed more motivation: "*I think it's great, and you feel more motivated when they give you an individual point*" (25B71-s); "*I agree and think that you are really motivated to do it better, to try to do it well*" (25G72-s). The continuous reinforcement system not only provides a greater probability of a conduct being repeated [71], but can also generate a feeling of desire (not one of liking) before being

given a point [72]; that is, when the learning task is performed: "*You feel motivated before being given a point*" (25G73-s). The teachers generally agreed with the designed points system: "*The points system is motivating*" (11B12-t); "*I thought that the gamified system was good because first, what is prioritized is that you improve. You have a baseline level and you obtain an end level. You have some results. You see what progress you make*" (6B13-t).

The points system can be motivating, but can also be perceived as an external control tool and a stressor [36, 66]. Although positive remarks were made about most cases, debate also took place in focus groups which, in relation to increased performance and making more efforts from being motivated by points, it was more overwhelming than other activities with less external feedback: "*I* [felt more tired] *with Just Dance Now because you have to concentrate more on the game and you had to do everything you saw on the screen. That's more overwhelming. So of course you felt more tired*" (26G56-s); "*Because in the other one, it was about how I invented dance steps. At the end, they were all repeated, and you felt calmer" (26B57-s); "You have more freedom to work*" (26G58-s). This could mean improved performance. Avoiding loss of points could be one of the gamifying focal points which, if badly managed, could lead to negative feelings and impulses [45]. Most gamifying dynamics [48], produced by making attempts by using the intervention (see Table 1), were positively valued by the students:

- Feeling of success: "*I thought that dancing and all that was great fun because you can meet your objectives*" (24G22-s).

- Feeling of progress: "*It was good because you could see everything you improved in*" (15G34-s).

- Feedback and reinforcement: "*green and yellow points told you that you'd improved and if you had done it well*" (15G34-s).

- Accumulability: "*It didn't matter if you forgot. But with Just Dance Now, as you get points, you lose points if you forget a part or stop moving your hand*" (26G59-s).

- Cooperation: "*Apart from feeling motivated to win points, you must also motivate your team so that it gets plenty of points*" (25G73-s).

- Feeling of challenge: "*I liked it because you felt motivated to work better, especially for double points* [challenges]. *I made the same effort, but felt more motivated to do it well*" (25G74-s).

- Competitiveness: "*Yes, you feel more motivated to be successful at it* [be the first]. *If you're not first you think «Well, I can manage it another week»*" (24B104-s).

- Position: "*I thought the classification was good because if I am too low, I can move up, and if I ´m in a good position, I can stay where I am*" (26G37-s).

However, no remark was made about self-expression and collectability. Feeling of challenge has previously been reported as a source of motivation, but also as a possible barrier if not adapted in several studies [29, 32, 73]. Feeling of success after meeting previously set objectives coincides with that found during interviews and observations [27, 67], where the students also indicated and remarked on the results to other friends and teachers, and they gave positive details of the game in line with this.

Boredom was not reported in the E group with time, unlike [27, 31, 74], where children lost interest in exergame. In our study, gamification seemed to produce constant fun and motivation as a feeling of progress being made was attempted using the gamifying board and levels, and also a feeling of challenge by daily and weekly challenges being set, and the level of difficulty progressively increased. Therefore, it would appear that gamification better maintained

high motivation with time *versus* exergame. This can be accounted for because what [74] recommended can be achieved with gamification for exergames; that is, sustainability, adaptability and sociability. However, boredom was reported in both groups for corporal expression activities, whenever they were performed without using exergame or gamification. Thus conducting future studies with other PE contents is recommended because contents could be a limiting factor, as discovered in [29].

The specific exergame effects can be related more to flow [75], dance enjoyment and an entertainment alternative; i.e., that which is directly linked to the presence and structure of *Just Dance Now*. Exergame directly increased dance enjoyment, or related negative values disappeared in other cases: "*I now like* [dancing] *much more. Before, I didn't feel like letting myself go. But when we started dancing with Just Dance Now, I had the chance to. . . express myself*" (16B13-s). One of the possible psychological effects it had was flow, specifically in relation to its characteristic feeling of time being transformed. This was apparently achieved more by the video game than by its gamified atmosphere in general: "*I ended up sweating with both* [dancing with and without exergame], *but I felt more tired with Just Dance Now*" (24G65-s); "*But you did not think about feeling tired with Just Dance Now*" (24G65-s). These remarks are coherent with the field notes, which indicated that students performed more PA with exergame. Feeling more tiredness can be explained by feeling more effective physical fatigue. More perceived fatigue can be explained by more physical effective fatigue, which is perhaps explained by making more effort and greater engagement which was, in turn, because the participants experienced more fun [27]. The partial effect on flow state could focus more on paying attention to the task and paying less attention to external elements, or even to the internal elements of one's own body. This can be explained by the capacity of exergames to lead to immersion [32], which facilitates a feeling of security and less anxiety [32, 76]. Each dance lasted 2–3 minutes on average, so immersion could be interrupted between one song and the next owing to limited Wi-fi network signals or to social interruptions as in [27]. Another specific effect of exergame was that it was considered a healthier entertainment alternative: "*I think this project is very good as it helps children to think who, apart from playing, many now play video games, although some children practice exercise while playing*" (25B4-s).

By considering what the OQQs and interviews found, gamification appeared to have conferred a greater general positive feeling (28.6% for gamification vs. 21.9% for exergame) and led to remarks about motivation in different gamifying dynamics made by all the students. Conversely, exergame had more specific effects, like fun and motor learning, and led to more expectations of use for some students.

It ought to be stated that a video game, the exergame *Just Dance Now* in our case, is an intensely gamified system with a highly specific set mechanics; that is, with a very limited and specific material design, which is why only certain people like them. Nevertheless, the intervention's didactic design has a more open gamifying mechanics based on proposals of didactic [41, 48], psychological [45, 75] and technological [44, 46] interests, and can be better adapted to different psychological profiles, and also to student preferences. Empirical results apparently back this differentiation between the exergame mechanics and the gamification mechanics.

## Conclusions

This study is the first to examine the applicability and usefulness of an intervention by combining gamification as a didactic method and an exergame as an educational resource. This research is an example of combining rigor in the didactic design of interventions and rigor of a qualitative methodology in a natural experiment.

The gamified intervention done with exergame, the object of this study, was partially applicable. The facilitators were the realism of its didactic design and its adaptability to different educational contexts. The main barriers were the necessary materials and facilities. The attitudes shown by teachers and students were very positive, but their expectations of its future use were inconclusive. The compatibility with the school curriculum was complete. This intervention can gain applicability through the suggestions proposed by all the participants. The usefulness of this intervention on the whole (gamification and exergame) has been proven to produce more fun, motivation, better liking dance, feeling less shame about dancing, more creative inspiration, more autonomous learning, and provides a digital leisure alternative. However, the intervention has not been shown to promote PEx in all students, according to the profile and previous habits associated with PEx and traditional video games, or to solve group work problems to any greater extent than traditional teaching.

Despite the difficulty of separating the effects of gamification and exergame, gamification provided a greater overall positive feeling and more motivation in most students. Exergame specifically produced more fun and motor learning. Although scientific interest has been shown in understanding separate effects, the combination of gamification as a method and exergame as a tool is considered significant in didactic terms. Our findings may mean that this study is one of the few to provide positive evidence for educational gamification. The *Mechanics-Dynamics-Aesthetics* gamification model and the *Just Dance Now* exergame may be applicable and useful for didactics in PE, but it is necessary to deal with the various detected aspects to help improve teaching interventions.

The implications of this study include understanding the potential construction of educative gamification and primary school didactics to build applicable and useful instructional environments, which would extend the useful reference framework.

## Strengths and limitations

Although some previous studies have studied gamification or exergame in educational systems, as far as are aware this is the first study to analyze the effects of an educative intervention that combines both phenomena. It is also the only study based on a natural experiment to use a C group, and is taught by the same teacher whose approach is taken from the qualitative methodology to allow an in-depth understanding of the effects. The key strengths of our study include triangulating: both the C and E intervention results; both teachers' and pupils' perspectives by considering their expectations and viewing interventions; the data from four different qualitative data collection techniques; both quantitative and qualitative data.

Interventions lasting 1 month (9 hours) were made in line with the traditional way of programming school contents [41]. However, due to logistic complexity, certain technical mismatches appeared which made the actual time used with the exergame slightly different among participants. Although the duration of studying the applicability sufficed, longitudinal studies are necessary to analyze the utility of such interventions in the long term. Another limit when interpretating this article is to use specific gamification elements and the concrete use of the *Just Dance Now* exergame. Thus future studies could include different gamification elements, another exergame, or distinct PE contents to be applied and compared.

A limitation of the present study is the unequal existence of female teachers and male teachers caused by prioritizing the choice of participating schools according to Criteria-Based Selection (predisposition, availability of material, etc.), and not the teachers' profiles. Future research designs could attempt to gather a more proportional number of teacher types. Moreover, it has been found that some students in this study played exergame at home while

participating in this study, which could affect the results. Therefore, it would be very interesting for future studies conducted from a multi-method approach to bear in mind these variables.

The applicability and utility of interventions based on gamification and exergames may change in participants of different ages [56]. Moreover, the video games phenomenon is associated with males [77], dance with females [61], and some gender differences appear in relation to the use of exergames [29, 33]. Therefore, it would be interesting to know the effects of such interventions according to gender as a future research line. Another interesting future research line would be to incorporate other indirect agents that participate in the school community, such as group tutors or parents.

## Supporting information

**S1 Material. Classification trees.**
(DOCX)

**S2 Material. Transcripts.**
(DOCX)

**S1 Checklist. Consolidated criteria for reporting qualitative studies (COREQ) checklist for qualitative data.**
(DOCX)

## Acknowledgments

The authors wish to thank all the schools, teachers and pupils for their involvement in this study.

## Author Contributions

**Conceptualization:** Alejandro Quintas-Hijós, Juan Carlos Bustamante.

**Formal analysis:** Alejandro Quintas-Hijós, Carlos Peñarrubia-Lozano.

**Funding acquisition:** Juan Carlos Bustamante.

**Investigation:** Alejandro Quintas-Hijós, Carlos Peñarrubia-Lozano.

**Methodology:** Alejandro Quintas-Hijós, Juan Carlos Bustamante.

**Project administration:** Juan Carlos Bustamante.

**Supervision:** Juan Carlos Bustamante.

**Validation:** Alejandro Quintas-Hijós, Carlos Peñarrubia-Lozano.

**Writing – original draft:** Alejandro Quintas-Hijós.

**Writing – review & editing:** Carlos Peñarrubia-Lozano, Juan Carlos Bustamante.

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
