## [Decision Letter · Decision Letter 0]

7 Jan 2020

PONE-D-19-32666

Analysis of the applicability and utility of a gamified didactics with exergames at primary schools: qualitative findings from a natural experiment

PLOS ONE

Dear Dr. Bustamante,

Thank you for submitting your manuscript to PLOS ONE. After careful consideration, we feel that it has merit but does not fully meet PLOS ONE’s publication criteria as it currently stands. Therefore, we invite you to submit a revised version of the manuscript that addresses the points raised during the review process.

We would appreciate receiving your revised manuscript by Feb 21 2020 11:59PM. To enhance the reproducibility of your results, we recommend that if applicable you deposit your laboratory protocols in protocols.io, where a protocol can be assigned its own identifier (DOI) such that it can be cited independently in the future. For instructions see: http://journals.plos.org/plosone/s/submission-guidelines#loc-laboratory-protocols

We look forward to receiving your revised manuscript.

Kind regards,

Francisco Javier Huertas-Delgado, Ph.D.

Academic Editor

PLOS ONE

Additional Editor Comments:

Dear authors, thank you very much for the work conduced. Please, take into special consideration all the reviewers' comment and specially please add more information in your discussion section.

Journal Requirements:

2. We note that Figures 1 and 2 include an image of participants in the study.

If you are unable to obtain consent from the subjects of the photograph, you will need to remove the figures and any other textual identifying information or case descriptions for this individual.

3. Please amend your manuscript to include your abstract after the title page.

Reviewers' comments:

Reviewer's Responses to Questions

**Comments to the Author**

1. Is the manuscript technically sound, and do the data support the conclusions?

Reviewer #1: Yes

Reviewer #2: Yes

Reviewer #3: Yes

2. Has the statistical analysis been performed appropriately and rigorously? 

Reviewer #1: Yes

Reviewer #2: Yes

Reviewer #3: Yes

3. Have the authors made all data underlying the findings in their manuscript fully available?

Reviewer #1: Yes

Reviewer #2: Yes

Reviewer #3: Yes

4. Is the manuscript presented in an intelligible fashion and written in standard English?

Reviewer #1: Yes

Reviewer #2: Yes

Reviewer #3: Yes

5. Review Comments to the Author

Reviewer #1: In this manuscript, researchers tried to use exergaming as a method of increasing motivation and participation of students in the Physical Education (PE). The researchers designed an appropriate method for examining their hypotheses about executing this research well.

It is just one issue that is as follow:

In line 126: write boys: (53.2% girls, n=222, 46.8% boys, n=195)

Reviewer #2: 70-73 Explain why focus groups are more appropriate for younger children; 75 Gender means sex? 33 (964) wrong citation 89-96 improve the presentation of research problems, explain the meaning of Why? 97-100 All objetives should be write with “to study, to understand” 119 the name of your selection is Criteria Based Selection 238 There is no Triangulation of people . You have triangulation of observers, researchers, sources or instruments

Reviewer #3: Please, find some comments and suggestions for the manuscript entitled: “Analysis of the applicability and utility of a gamified didactics with exergames at primary schools: qualitative findings from a natural experiment. I would like to congratulate Authors for this interesting study. Nevertheless, I consider there are some changes that should be done.

Page 3, line 66: Please, add an abbreviation for the physical activity (PA).

Page 4, line 74: It should be "in elementary students" or "in primary students"? Please, double-check.

Page 4, line 85: Studies for me means more than one. Please, add some appropriate references.

Page 4, line 87: “in primary schools” instead of “in schools”

Page 5, line 102: Based on the aims, did Authors generate any official hypotheses for this work? If so, it will be helpful to include in the Introduction.

Page 5, line 120: “primary schools” instead of “schools”

Page 6, lines 125-126: Please, give more detailed information about the teachers who took part in your study (male/female, work experience, age).

Page 6, lines 127-128: “Year 6” and “Year 7” – this is unclear for me. Maybe it will be better to use “Grade”? Also, please add detailed information about students’ sex (how many girls and boys were in each grade).

Page 6, line 141, line 143 and line 145: The term "treatment" is not the most appropriate one. I suggest replacing that word for a different one.

Page 6 and page 7, lines 145-146: Each treatment lasted 9 hours or involved 12 sessions, which was applied to each school for 4 or 6 weeks – this is unclear to me. Also, please explain, why did you choose 1 month (9 hours) of the intervention? What was the reason?

Page 7, line 148: Why only male researcher-teacher? Also, please explain what means researcher-teachers.

Page 7, line 152: Could you explain what means “dancing level 8”?

Page 8, line 717: I suggest to remove the horizontal lines (even some of them), however, it should stick to the Journal style (it refers to all tables in the manuscript). Also, try to be consistent – sometimes there is a period, sometimes a comma, sometimes a small then capital letter. In general, I suggest using a smaller font size in all tables.

Page 10, lines 196-199: When the students had to enter the score? Just after the PE classes or later? Who was responsible for helping children during that stage?

Pages 10-11, lines 207, 211, 219, 228: I suggest to use numbers instead of dashes to indicate each qualitative measures.

Page 11, line 230: Because only certain students (and teachers? - this is unclear) were interview, then it will be good add some information about the selection criteria.

Page 12, line 243: What means semi-private school? Don’t you think that the type of school may differentiate the results?

Pages 14, line 284: I suggest to use the abbreviation for control (C) and experimental (E) in the section “Results and Discussion”.

Page 19, line 315: Probably it should be “B” for boys and G for girls (capital letters).In general, it looks that in the project were involved more male teachers than male. Please, explain why.

Page 23, line 421: PA instead of physical activity. Please, change it.

Page 25, line 486: The same as above.

Page 27, line 516: Several studies means more than two for me. Please add more references here.

Page 30, line 608: What “N” stand for?

Page 33, lines 673-674: Many students indicated that they liked exergame and they downloaded it at home. Do not you think that might differentiate the results (increase them)?

Page 37, line 762: Several studies for me means more than two. Please add some references here.

Page 38, line 788: PA instead of physical activity. Please, change it.

Page 41, line 877: I would like to congratulate Authors because the references are very updated, and the information very new.

However, for bibliographic references, criteria must be unified (for example – please see: page 43, lines 924-925 or page 45 lines 964-966). Please, double-check the References section and follow the Journal style.

6. PLOS authors have the option to publish the peer review history of their article (what does this mean?). If published, this will include your full peer review and any attached files.

Reviewer #1: No

Reviewer #2: Yes: FERNANDO Barragán MEDERO

Reviewer #3: No

---

## [Author Response · Author response to Decision Letter 0]

21 Feb 2020

Dear Editor,

It is a pleasure for us to receive the opportunity to improve our manuscript by taking into account the reviewers’ interesting comments. We expect this revision will allow us to make significant progress in the present form of our paper, and it provides us with the opportunity for it to be considered for publication.

We would like to thank the reviewers for their detailed comments and helpful suggestions. Below we have listed our replies to each comment and suggestion. The changes have been highlighted and included in the manuscript.

Response to Editor

Additional Editor comments. Dear authors, thank you very much for the work conduced. Please, take into special consideration all the reviewers' comment and specially please add more information in your discussion section.

Response: We added more information in the discussion to address some of the reviewers’ comments.

Response to Reviewers

Reviewer #1. In line 126: write boys: (53.2% girls, n=222, 46.8% boys, n=195)

Response: We have corrected the error.

Reviewer #2. 70-73 Explain why focus groups are more appropriate for younger children.

Response: We stated that “Focus groups are especially appropriate when seeking children’s”. We have explained this statement better, based on the study by Kennedy, Kools and Krueger (38), to make it clearer to readers. Three reasons are given: they imply a shared intervention experience, which allows to take children’s spontaneity; adults have a better chance of understanding what children have to say in their interaction and behavior context; focus groups free children and researchers of the data collection limitations placed by literacy/reading levels.

Reviewer #2. 75 Gender means sex?

Response: The term "gender" was used as in a previous study (De Vet, Simons and Wesselman, 2012; 31) because we considered it was broader as it does not refer only to physical, but also to psychological and social aspects, and was more accurate for our study. Now in the data collection we only ask the participants to consider the dichotomous variable Man-Woman.

Reviewer #2. 33 (964) wrong citation

Response: The error has been corrected.

Reviewer #2. 89-96 improve the presentation of research problems, explain the meaning of Why?

Response: Indeed, "why" was it an ambiguous question as it has been written in that context. It has been replaced with a more specific question, which is really answered in the study: "What are the reasons that make exergaming and gamification more applicable and useful?"

Reviewer #2. 97-100 All objectives should be write with "to study, to understand"

Response: The modifications have been applied.

Reviewer #2. 119 the name of your selection is Criteria Based Selection

Response: The name of the selection has been added to give more information to readers.

Reviewer #2. 238 There is no Triangulation of people . You have triangulation of observers, researchers, sources or instruments

Response: We have replaced "triangulation of people" with "triangulation of observers", and “triangulation of techniques” with “triangulation of instruments”, to confer adequacy. We have also added triangulation of researchers, as the reviewer suggests. 

Reviewer #3. Page 3, line 66: Please, add an abbreviation for the physical activity (PA).

Response: We have added the abbreviation. 

Reviewer #3. Page 4, line 74: It should be "in elementary students" or "in primary students"? Please, double-check.

Response: We have opted for "in elementary students" rather than "in primary students" to make the whole text consistent and because American English is used.

Reviewer #3. Page 4, line 85: Studies for me means more than one. Please, add some appropriate references.

Response: We have added several appropriate studies on gamification and exergaming to better support the statement.

Reviewer #3. Page 4, line 87: "in primary schools" instead of "in schools"

Response: We have added "in elementary" to clarify and make the hole text consistent.

Reviewer #3. Page 5, line 102: Based on the aims, did Authors generate any official hypotheses for this work? If so, it will be helpful to include in the Introduction.

Response: It is a good appreciation. However, we do not use any hypothesis because, by starting with a qualitative approach, our interest focused on understanding the phenomenon of interest through the involved actors to learn more about the studied reality.

Reviewer #3. Page 5, line 120: "primary schools" instead of "schools"

Response: We have added "in elementary" to clarify and make the hole text consistent.

Reviewer #3. Page 6, lines 125-126: Please, give more detailed information about the teachers who took part in your study (male/female, work experience, age).

Response: That detailed information has been added to the manuscript.

Reviewer #3. Page 6, lines 127-128: "Year 6" and "Year 7" – this is unclear for me. Maybe it will be better to use "Grade"? Also, please add detailed information about students' sex (how many girls and boys were in each grade).

Response: Our basis was the nomenclature of the United Kingdom ("Year"), but it is ambiguous. Therefore, we have added the term "grade" to refer to a more international system. Perhaps, what is always clear is children’s ages. We have specified information about how many students of each gender belonged to each “grade”.

Reviewer #3. Page 6, line 141, line 143 and line 145: The term "treatment" is not the most appropriate one. I suggest replacing that word for a different one.

Response: Following the reviewer’s suggestion, we have replaced the term "treatment" with “condition”, in the sense of a "different educational situation". Moreover, we have replaced "treatment" with "intervention" in some cases to better reflect the nature of the design and to apply the "condition" that was performed.

Reviewer #3. Page 6 and page 7, lines 145-146: Each treatment lasted 9 hours or involved 12 sessions, which was applied to each school for 4 or 6 weeks – this is unclear to me. Also, please explain, why did you choose 1 month (9 hours) of the intervention? What was the reason?

Response: Each didactic intervention should last a minimum of 4 weeks because the PRE-test was applied 1 week before and 1 week later. In this way, the minimum duration between the PRE-test and the POST-test of 40 days was achieved (as advised by Rey, Espinosa and Hidalgo, 1989, and by Fontes and Fontes, 2016). Moreover, the usual way to schedule school physical education classes in Spain is usually 1 month (Larraz, 2012). So this research has been adapted to the Spanish context. However, some Spanish primary schools have more shorter sessions or fewer longer sessions, but they all have 135 minutes per week (or 9 h each month). To balance this variance, the study agreed that all the children would receive a 9-hour teaching unit, regardless of class duration (45 minutes or 1 hour), and also regardless of the number of weeks (between 4 and 6 weeks as some holidays take place in different months). These methodological clarifications have been added to the manuscript.

Reviewer #3. Page 7, line 148: Why only male researcher-teacher? Also, please explain what means researcher-teachers.

Response: "Researcher-teacher" refers to the person who played the role of researcher in this study, who also taught classes in schools. Priority has been given to the fact that it was always the same person who taught the teaching units to all the children in this study, who was male in this case This methodological decision was made to avoid possible effects on the responses by participants due to the variety of teachers, and in order to compare all the schools and the two groups. These methodological clarifications have been added to the manuscript.

Reviewer #3. Page 7, line 152: Could you explain what means "dancing level 8"?

Response: Here it refers to the exergame dance with a level of difficulty of 8 out of 10. However, we had to remove this text due to the Journal’s data protection policy associated with the shown figure.

Reviewer #3. Page 8, line 717: I suggest to remove the horizontal lines (even some of them), however, it should stick to the Journal style (it refers to all tables in the manuscript). Also, try to be consistent – sometimes there is a period, sometimes a comma, sometimes a small then capital letter. In general, I suggest using a smaller font size in all tables.

Response: We have made the table’s typography consistent and have removed some horizontal lines. Some have been maintained so that readers can see the association among the elements of mechanics, dynamics and aesthetics in this study.

Reviewer #3. Page 10, lines 196-199: When the students had to enter the score? Just after the PE classes or later? Who was responsible for helping children during that stage?

Response: The students entered the points obtained in the shared computer immediately after dancing each dance in class time. The researcher-teacher organized a few minutes for this task, and helped with management, especially during the first sessions. We have clarified this issue in the manuscript.

Reviewer #3. Pages 10-11, lines 207, 211, 219, 228: I suggest to use numbers instead of dashes to indicate each qualitative measures.

Response: Dashes have been replaced with numbers to gain clarity. The changes made in Table 2 have also been applied. 

Reviewer #3. Page 11, line 230: Because only certain students (and teachers? - this is unclear) were interview, then it will be good add some information about the selection criteria.

Response: All the teachers associated with each group of students were interviewed, which has been better specified in the text. The students' choice of selecting children was made by the researcher-teacher according to his/her experience throughout the intervention if it was observed that a student's profile could be very interesting: two students were chosen for showing special interest in classes and increasingly throughout the didactic unit, and two students were chosen for quite the opposite. These specifications have been added to the manuscript.

Reviewer #3. Page 12, line 243: What means semi-private school? Don't you think that the type of school may differentiate the results?

Response: The "Semi-private" school (or "semi-public”) is a very common type of school in Spain. It refers to a privately managed school, but which partially receives public money. As public and “semi-private” schools are very frequent, a decision was made to take into account two schools of each type in this study because they could affect the results. We have added this explanation about the Spanish context. The public or semi-private character of the schools in this study did not seem to determine differences in our study. Nevertheless, we have added some considerations about this issue in the "Logistics" section of the Discussion.

Reviewer #3. Pages 14, line 284: I suggest to use the abbreviation for control (C) and experimental (E) in the section "Results and Discussion".

Response: Those abbreviations have been added whenever they appear in the "Results and Discussion" Section.

Reviewer #3. Page 19, line 315: Probably it should be "B" for boys and G for girls (capital letters).In general, it looks that in the project were involved more male teachers than male. Please, explain why.

Response: Capital letters have been added throughout the "Results and Discussion" section. 

As Table 3 indicates, there are two female teachers and six male teachers, which is a 25%-to-75% ratio. This was due to having prioritized the choice of participating schools according to the Criteria-Based Selection (predisposition, availability of material, etc.), and not to teachers’ profiles. Thus we have added the unequal distribution of teachers as a study limitation. 

Reviewer #3. Page 23, line 421: PA instead of physical activity. Please, change it.

Response: The change has been applied, and is now consistent with the rest of the text.

Reviewer #3. Page 25, line 486: The same as above.

Response: The change has been applied.

Reviewer #3. Page 27, line 516: Several studies means more than two for me. Please add more references here.

Response: We have added several studies which have qualitatively found similar attitudes of PE teachers.

Reviewer #3. Page 30, line 608: What "N" stand for?

Response: The “N” did not form part of the code. This was an error that has been corrected.

Reviewer #3. Page 33, lines 673-674: Many students indicated that they liked exergame and they downloaded it at home. Do not you think that might differentiate the results (increase them)?

Response: The reviewer's assessment is quite correct given that this variable could affect the results. Therefore, we have included this issue in the "Strengths and limitations" section as an important element to consider in future research lines.

Reviewer #3. Page 37, line 762: Several studies for me means more than two. Please add some references here.

Response: The statement has been modified to be consistent, and three studies have been referenced.

Reviewer #3. Page 38, line 788: PA instead of physical activity. Please, change it.

Response: The change has been applied.

Reviewer #3. Page 41, line 877: I would like to congratulate Authors because the references are very updated, and the information very new.

However, for bibliographic references, criteria must be unified (for example – please see: page 43, lines 924-925 or page 45 lines 964-966). Please, double-check the References section and follow the Journal style.

Response: We appreciate this indication. All the bibliographic references in the text have been reviewed and corrected.

As you can see, we have considered all the advice offered and concerns raised by the reviewers. We believe that the reviewers are right and have given us good advice to improve our manuscript.

We hope that you receive our change proposals optimistically with a view to publishing this manuscript.

Yours faithfully,

The authors.

---

## [Decision Letter · Decision Letter 1]

20 Mar 2020

Analysis of the applicability and utility of a gamified didactics with exergames at primary schools: qualitative findings from a natural experiment

PONE-D-19-32666R1

Dear Dr. Bustamante,

We are pleased to inform you that your manuscript has been judged scientifically suitable for publication and will be formally accepted for publication once it complies with all outstanding technical requirements.

With kind regards,

Francisco Javier Huertas-Delgado, Ph.D.

Academic Editor

PLOS ONE

Additional Editor Comments (optional):

Reviewers' comments:

Reviewer's Responses to Questions

**Comments to the Author**

1. If the authors have adequately addressed your comments raised in a previous round of review and you feel that this manuscript is now acceptable for publication, you may indicate that here to bypass the “Comments to the Author” section, enter your conflict of interest statement in the “Confidential to Editor” section, and submit your "Accept" recommendation.

Reviewer #1: All comments have been addressed

Reviewer #2: All comments have been addressed

Reviewer #3: All comments have been addressed

2. Is the manuscript technically sound, and do the data support the conclusions?

Reviewer #1: Yes

Reviewer #2: Yes

Reviewer #3: Yes

3. Has the statistical analysis been performed appropriately and rigorously? 

Reviewer #1: Yes

Reviewer #2: Yes

Reviewer #3: Yes

4. Have the authors made all data underlying the findings in their manuscript fully available?

Reviewer #1: Yes

Reviewer #2: Yes

Reviewer #3: Yes

5. Is the manuscript presented in an intelligible fashion and written in standard English?

Reviewer #1: Yes

Reviewer #2: Yes

Reviewer #3: Yes

6. Review Comments to the Author

Reviewer #1: (No Response)

Reviewer #2: I accept all changes made in a correct way. The article has improved in a deep way with the nes explanations done.

Reviewer #3: (No Response)

7. PLOS authors have the option to publish the peer review history of their article (what does this mean?). If published, this will include your full peer review and any attached files.

Reviewer #1: Yes: Ali Abbasi

Reviewer #2: No

Reviewer #3: No

---

## [Editor Report · Acceptance letter]

25 Mar 2020

PONE-D-19-32666R1 

Analysis of the applicability and utility of a gamified didactics with exergames at primary schools: qualitative findings from a natural experiment 

Dear Dr. Bustamante:

I am pleased to inform you that your manuscript has been deemed suitable for publication in PLOS ONE. Congratulations! Your manuscript is now with our production department. 

With kind regards,

on behalf of

Dr. Francisco Javier Huertas-Delgado 

Academic Editor

PLOS ONE